# Towards the Provision of Palliative Care Services in the Intensive Coronary Care Units: Nurses’ Knowledge, Training Needs, and Related-Barriers

**DOI:** 10.3390/healthcare11121781

**Published:** 2023-06-16

**Authors:** Baraa Abu-Aziz, Areefa S. M. Alkasseh, Jonathan Bayuo, Hammoda Abu-Odah

**Affiliations:** 1Nasser Medical Complex Hospital, Ministry of Health, Gaza P.O. Box P860, Palestine; baraa-salama-1993@outlook.com; 2Department of Midwifery, Nursing College, Islamic University of Gaza, Gaza P.O. Box P108, Palestine; abahri@iugaza.edu.ps; 3School of Nursing, The Hong Kong Polytechnic University, Hong Kong 999077, China; jonathan.bayuo@connect.polyu.hk; 4WHO Collaborating Centre for Community Health Services (WHOCC), School of Nursing, The Hong Kong Polytechnic University, Hong Kong 999077, China; 5Nursing and Health Sciences Department, University College of Applied Sciences (UCAS), Gaza P.O. Box P860, Palestine

**Keywords:** cross-sectional study, coronary diseases, intensive care, knowledge, Palestine, palliative care, training

## Abstract

Despite the notable benefits of palliative care (PC) for patients with chronic diseases, its delivery to people with cardiac problems, particularly in the Middle East region (EMR), remains a critical issue. There is a scarcity of research assessing nursing staff’s needs and knowledge in providing PC to cardiac patients in the EMR. This study aimed to assess the level of knowledge and needs of PC among nurses towards the provision of PC in intensive coronary care units (ICCUs) in the Gaza Strip, Palestine. It also identified the barriers to the provision of PC services in ICCUs in the Gaza Strip. A hospital-based descriptive quantitative cross-sectional design was adopted to collect data from 85 nurses working in ICCUs at four main hospitals in the Gaza Strip. Knowledge about PC was collected using a developed questionnaire based on the Palliative Care Quiz Nursing Scale (PCQN) and Palliative Care Knowledge Test (PCKT). PC training needs and barriers were assessed using the PC Needs Assessment instrument. Approximately two-thirds of nurses did not receive any PC educational or training programs, which contributed to their lack of PC knowledge. Most nurses would like to enroll in PC training programs, such as family support and communications skills courses. Nurses reported that there was a high demand for PC guidelines and discharge planning for patients with chronic illnesses. Insufficient healthcare professionals’ knowledge about PC and a staff shortage were the main barriers to integrating PC into the Gaza healthcare system. This study suggests incorporating PC into nursing curricula and in-service training, and it covers both basic and advanced PC principles. Intensive coronary care unit nurses need knowledge and training about PC, guidance, and support to provide appropriate care to patients with cardiovascular issues.

## 1. Introduction

Palliative care (PC) is a holistic approach to care for patients with life-limiting illnesses and/or suffering, such as heart failure [1]. It tries to reduce pain, lengthen the patient’s life, and improve the quality of life (QOL) for the patient and their family [1]. The demand for PC services has expanded due to global demographic changes and the increasing incidence of non-communicable diseases (NCDs) [2]. The World Health Organization (WHO) recognized PC as essential for comprehensive non-NCD services [3].

Cardiovascular diseases (CVDs) are one of the NCDs and the main cause of death worldwide, accounting for an estimated 17.9 million deaths in 2019 alone [4]. Over 70% of all deaths occur in lower- and middle-income countries (LMICs) [4]. In Palestine, CVDs continue to top the cause of mortality list, with a prevalence estimated at 24.7% [5]. The disease trajectory of most CVDs warrants the inclusion of PC services either at the acute or end-of-life stages. Despite the notable benefits of PC in CVDs (particularly heart failure) and the enormous burden of CVDs, the utilization of PC to improve the health of cardiac disease patients remains low [6].

Palliative cardiovascular care is essential for figuring out the scope and timing of medical management plans and coordinating care for symptoms based on the patient’s values, wishes, and preferences [7]. It focuses on decreasing physical, psychological, social, and spiritual pain and is concerned about symptom management and family support [7]. The early integration of PC in cardiac patients can significantly improve the QOL, satisfaction, and symptoms of patients and their family members [8]. However, PC remains underutilized for managing patients with cardiac diseases, particularly in LMICs [9]. There are numerous reasons for the underutilization of PC in LMICs, with the shortage and unpreparedness of healthcare professionals being the primary ones [10].

Nurses are the most prominent healthcare professional group and play a key role in PC. They coordinate care for patients, their families, and other healthcare workers [11]. They are responsible for practically all bedside care; therefore, they should have adequate knowledge and training in PC to fulfil the demands of the patients [12]. Appropriate knowledge and training also aid nurses in efficiently communicating with patients and their families and easing their suffering [12,13].

Few studies have examined PC for cardiac patients; instead, the majority have examined the attitudes of medical professionals. For instance, Ziehm et al. [14] conducted an interview study to analyze healthcare professionals’ perspectives regarding PC for patients with chronic heart failure and found that PC for chronic heart failure was a neglected topic in research and practice that needed more attention. They also reported that a lack of PC knowledge and shortcomings in communication and cooperation among different professional groups were the main barriers to PC provision. Kim and Hwang [15] assessed the knowledge and attitudes of nurses towards the care of dying patients with heart failure in South Korea. They found insufficient knowledge and poor attitudes towards the care of the dying. Even though the previous studies’ findings were significant, no research has been conducted to determine the nursing needs for PC in patients with CVDs. In order to assess the needs of nurses working in intensive coronary care units (ICCU), where PC services are unavailable and not integrated into the healthcare system, this study was carried out in the Gaza Strip, Palestine. 

In the Gaza Strip, efforts have been made to integrate PC into the health system; however, the introduction has encountered various obstacles, including the absence of policies to support PC and the subordinate status of PC compared to acute care [16]. As evidenced by the Palestinian Strategic Plan 2021–2025, the Ministry of Health (MOH) has recently directed the introduction of PC into the health care system [16] to determine the nursing care needs and provide solid and dependable data on the care provided to cardiac patients, which may be used for strategic planning towards developing PC in ICCUs at government hospitals in the Gaza Strip. This study’s results will contribute to improving healthcare services provided to the cardiac patients and will aid the MOH in developing a strategic plan in the coming years. This study aimed to assess the level of knowledge and needs of palliative care (PC) among nurses towards the provision of PC in intensive coronary care units in the Gaza Strip as well as identify the barriers to the provision of PC services in ICCUs. In addition, it aimed to examine the relationships between nurses’ demographic and professional variables and their knowledge about PC. Our null hypothesis posited that no associations regarding all demographic and professional variables and nurses’ knowledge about PC would be found.

## 2. Materials and Methods

### 2.1. Study Design

A hospital-based descriptive cross-sectional quantitative design was applied in this study. The study was conducted in ICCUs at the five main governmental hospitals in the Gaza Strip (European Gaza Hospital, Nasser Medical Complex, Indonesian Hospital, Al Shifa Medical Complex and Al Aqsa Hospital), where intensive coronary care services are provided to patients with CVDs. The study was approved by the Islamic University of Gaza Ethics Committee (Ref No: 07/2021/GSGH) and by the Palestinian Health Research Council (protocol code PHRC/HC/972/21 in 11 October 2021). All nurses gave their informed written consent prior to being enrolled in this study. The SURGE (Survey Reporting Guideline) was used in reporting the current study [17].

### 2.2. Sample Calculation and Participant Selection

A convenience sampling method was adopted to recruit nurses from the five main governmental hospitals in the Gaza Strip. Nurses were eligible if they (i) were working in ICCUs at one of the five aforementioned governmental hospitals and (ii) were working at ICCUs for more than 6 months. Nurses were excluded if they worked at other hospitals that did not provide ICCU services to patients. Volunteers, students, and those with fewer than six months of ICCU experience were also barred from participating. 

In total, there were 105 nurses working at the aforementioned hospitals. The sample was calculated using the Epi-Info-7 StatCalc program with the following assumptions: a statistical power of 80%, a confidence level of 95%, an expected frequency/prevalence of 50%, and a margin of error of 5%. A minimum sample size of 82 participants was included. In this study, 85 nurses participated and were willing to complete the survey. 

### 2.3. Recruitment Procedures

From December 2021 to March 2022, the participants were recruited from the five main governmental hospitals in the Gaza Strip after obtaining the required ethical approval from the competent institutions. Two student volunteer nurses, whom the first author knows, were contacted for help in the distribution of the questionnaires and the processing of the data collection. The first author delivered the questionnaire pack to the two student nurses and clarified the eligibility criteria for study participants. Both students distributed the pack to the eligible nurses in the five hospitals. Each pack included a cover letter with a detailed description of the study, written informed consents, and the questionnaires. The student nurses also informed the participants about the box, which was placed on the counter of each department with the purpose of collecting the pack. The box was locked, and only the student nurses could open it to collect the returned pack.

### 2.4. Questionnaires Description

#### 2.4.1. Primary Outcome

##### Knowledge about PC

A self-administered questionnaire was developed based on previous literature reviews [1,12,18,19,20]. The questionnaire consists of 15 items designed to assess nurses’ PC knowledge and learning needs. Many of the items were based on the Palliative Care Quiz Nursing Scale (PCQN) [19] and the Palliative Care Knowledge Test (PCKT) [18]. Some items from PCQN and PCKT were excluded based on health experts’ panel opinions because they were not practically applicable to the Palestinian context [12]. From the Palliative Care Knowledge Test (PCKT), some of the items related to pain (item 4 and 6), dyspnea (item 9–12), psychiatric problems (item 13 and 14), and gastrointestinal problems (item 17–20) domains were excluded. For instance, item 4 in the pain domain entitled “When cancer pain is mild, pentazocine should be used more often than an opioid” was excluded because pentazocine was not utilized in Gaza hospitals. On the other hand, the panel recommended adding some items, (1) including “PC is exclusively for people who are in the last six months of life” [20], (2) and “PC is specifically for people with cancer” [20]. Thus, a total of 15 items were adopted to assess knowledge. The overall score of the questionnaire ranges from 0 (“lowest level of knowledge”) to 15 (“highest level of knowledge”). The final answers were coded 1 = correct, 0 = incorrect/do not know. The score out of 15 was converted to a percentage, and a higher percentage reflects good knowledge about PC. The internal consistency of the questionnaire was 0.813, indicating a good level of reliability.

#### 2.4.2. Secondary Outcomes

##### Palliative Care Training Needs and Barriers

The training needs and barriers to providing PC to ICCUs were assessed using the PC Needs Assessment instrument [21]. It is a comprehensive instrument covering the following areas: PC services available; barriers to PC provision and service delivery; available resources; populations requiring assistance; educational program topics attended in the past two years; and preferred learning methods.

##### Sociodemographic and Professional Variables

Sociodemographic and professional variables included the nurses’ gender, age, level of education, marital status, position, total years of working experience, years of experience in the ICCU, and history of PC education.

### 2.5. Statistical Analysis

The Statistical Package for Social Science (SPSS) version 25 program was used for data entry and analysis. Descriptive statistics were calculated to summarize the participants’ characteristics and the scales. All variables with *p* ≤ 0.05 in univariate analysis were selected for a multiple linear regression, which was used to predict the variables associated with the nurses’ knowledge of PC. All statistical tests were two-tailed, and *p*-values of less than 0.05 were treated as significant. 

## 3. Results

### 3.1. Nurses Characteristics

Of the 89 nurses invited to participate in this study, 85 agreed and completed the questionnaires, with a response rate of 95.5%. No missing data were reported in the questionnaires that were filled out by the study participants. More than half of the nurses (*n* = 49 (57.6%)) were male, and their mean age was 31.3 (SD = 6.5). Almost all nurses (*n* = 72 (84.7%)) held a bachelor’s degree. Only 33 nurses (38.8%) had previous PC education. Detailed descriptive statistics for the participants are presented in Table 1.

### 3.2. Knowledge of Nurses about PC

The overall knowledge mean score was 6.21 ± 1.97, with a range of 0 to 15, and the correct average rate was 41.4%, indicating that nurses lacked PC knowledge (Table 2). For nurses’ responses towards knowledge items, their correctly answered items ranged from 17.6% to 76.5%. The detailed responses of nurses to PC knowledge items are presented in Table 2. 

### 3.3. Palliative Care Training Needs 

Findings underscored that only a quarter of nurses (*n* = 24 (28.2%)) reported having attended training related to PC. For instance, 24.7% of nurses participated in pain and symptom management programs. Only 22.4% had participated in ethics issues at the end-of-life program (Table 3). Nevertheless, the majority of nurses wished to join PC training programs. For example, 82.4% would like to take family support courses, and 80.0% wish to enroll in communication skills courses. Furthermore, 78.8% would like to take psychological courses for dying patients, and the same percentage would like to join spiritual courses to help patients in need of PC (Table 3).

Nurses identified various preferred methods to learn more about PC; approximately 82.4% of nurses preferred partnerships with academic institutions to take training courses in PC. In addition, about 81.40% of nurses liked attending professional meetings or workshops; 77.6% preferred web-based learning, and 76.20% chose to partner with local hospice or home care providers.

### 3.4. Palliative Care Services Should Be Available in Hospital

As reported by the study participants, PC guidelines (*n* = 72 (85.0%)), discharge planning for patients with chronic illness (*n* = 71 (84.0%)), and quality improvement for the pain management program (*n* = 71 (84.0%)) were the most common PC services that should be available at hospitals prior to the integration of PC services in ICCUs. A detailed description of common PC services that should be available is presented in Table 4.

### 3.5. Barriers Hindering the Integration of PC

The study participants reported several barriers that hindered the integration of PC services into ICCUs. Lack of professionals’ knowledge about PC (*n* = 62 (72.9%)) and workforce shortages (*n* = 57 (67.1%)) were the main challenges to integrating PC into ICCUs. Inappropriate physical infrastructure for PCs, such as beds (*n* = 51 (60.0%)), was another barrier to PC provision. Finally, the invariability of policies on PC (*n* = 47 (55.3%)) was also considered a significant barrier. The common barriers are presented in Table 5.

### 3.6. Factors Associated on Nurses’ Knowledge about PC

The findings in the multiple linear regression showed that all personal and professional variables were not associated with nurses’ knowledge of PC (Table 6).

## 4. Discussion

This is the first study conducted in the Gaza Strip (Palestine) that assessed nurses’ PC needs in ICCUs. Palestine is categorized by the WHO in Group 3a, “Isolated PC Provisions” countries with PC activities, but it is fragmented and not adequately supported because of insufficient resources, donor dependence, and inadequate morphine [22]. Despite these initiatives, the majority of Palestinian nurses working in ICCUs did not participate in any PC education or training programs, contributing to their insufficient knowledge about PC. This result is consistent with previous studies conducted in Palestine [12], Jordan [23] and Egypt [24]. The insufficient knowledge among nurses in Palestine and neighboring countries might be attributed to the invariability of PC services in the healthcare system. Lack of PC training might play a role in the insufficient knowledge of nurses about PC [25]. Our findings showed that the majority of ICCU nurses did not receive any training in PC. This result is congruent with the Abu-Odah et al. study, which reported that only 9.5% of Gazan nurses had previous training on PC [12]. Within the broader global context, critical care or intensive care units have traditionally been considered non-PC settings, which may affect the inclusion of healthcare professionals working in these places as being excluded from PC training. However, as the need to integrate PC across all clinical units’ advances, healthcare staff in the critical care units are likely to benefit from ongoing staff education. In-service PC education programs are therefore needed to enhance nurses’ knowledge and skills about PC.

The majority of nurses are interested in enrolling in PC training programs, especially family support and communication courses. It is essential to reduce a patient’s family’s uncertainty regarding their loved one’s prognosis by providing them with sufficient and appropriate information [26]. Therefore, it is vital to provide nurses with family support skills in order for them to appropriately answer frequently asked questions about their patients’ health. Communication skills have been recognized as a cornerstone of quality care [27], and the benefits to patients and their families have been extensively demonstrated [27]. Discussing and communicating with patients and their families increases the chance of concordance between patient-reported goals of care and those noted in their medical records, as well as patient satisfaction with care. Therefore, it is vital to offer staff communication programs on how to care for and communicate with patients and their families about topics related to PC.

Palliative care guidelines at ICCUs, which were identified as the most common PC service by nurses in this study, should be offered in Gaza hospitals. Guidelines help nurses learn how to perform procedures and provide the best care to patients [16]. PC guidelines aid in improving nurses’ knowledge, attitudes, and practice in a positive way [24]. Thus, PC guidelines should be available and accessible in ICCUs to keep nurses’ knowledge up to date, improve practice, and foster a good attitude towards PC. This may not be only an issue in the Gaza Strip, as the notion of PC in critical care units remains an ongoing issue. It has been previously thought that PC in the ICU or even in the emergency department is an oxymoron considering the perceived conflicting underlying philosophies of these specialties. Thus, PC guidelines specific to critical care units are generally lacking globally. This need for PC guidelines specific to the critical care unit should therefore serve as a clarion call not only to Gaza, but also to the global healthcare community to work towards developing and implementing context-specific guidelines to improve PC uptake in the ICUs.

According to our nurses, the main barrier to integrating palliative care into ICCUs in the Gaza Strip was a lack of professional knowledge about PC. The study’s results are consistent with those reported in previous studies [10,12,16,28]. The lack of knowledge about PC among nurses in the Gaza Strip was attributed to the unavailability of PC in the healthcare system. It was also linked to a lack of PC curriculum in universities. The study elucidated the importance of developing PC education and training programs into university curricula and in-service training, and that they should address both basic and advanced PC principles.

Shortages of workforce are another barrier to the integration of PC into ICCUs in the Gaza Strip. Several studies have indicated that shortages of staff are the main barrier to PC provision [10,12,16]. For instance, Abu-Odah et al. [12] reported that a shortage of nursing staff is the main healthcare system challenge in the Gaza Strip. The Palestinian MOH should overcome the staff shortage by hiring new nurses to deal with patients. We acknowledge the financial constraints that the Gaza government faces; therefore, the government should cooperate with international health organizations to financially support the recruitment of new nurses.

Multivariate analysis revealed no association between nurses’ knowledge of PC and their personal and professional characteristics. The absence of an association between nurses’ knowledge and their personal and professional characteristics has also been observed by other studies. However, Palestinian studies showed a positive association between PC knowledge and a high educational level and previous training [12,29]. There was no association in this study because the majority of nurses reported a lack of PC knowledge and did not attend any PC training. The homogeneity of nurses’ characteristics in the Gaza Strip suggests implementing PC education and training for all nurses to improve their knowledge, despite their age, education level, and years of working experience.

This study has a number of limitations, one of which is that it adopts a cross-sectional design, which makes it difficult to examine the causal relationship between the different factors. The limited number of participants in the study makes it difficult for us to generalize the results. Considering that the population in this study is quite small, the risk of background bias may occur, which may affect the accuracy and representativeness of the findings. Additionally, the study focused solely on nurses, and its findings may not reflect the views of all healthcare professionals in the Gaza Strip. In addition, the exclusion of some items in the PCQN and PCKT instruments based on Palestinian experts’ panel opinions makes it difficult to compare our study findings with those of other studies in different populations. Furthermore, the lack of a pilot test makes it difficult to determine the validity of the study instruments and identify their weakness. However, this is the first study to be undertaken in the Gaza Strip to examine the needs of nurses for PC in ICCUs. The findings of this study will help enhance healthcare services in the Gaza Strip.

## 5. Conclusions

This study has demonstrated insufficient PC knowledge among Palestinian nurses working in ICCUs. This study also revealed that most nurses had a great desire to learn more about PC. The study underscored that insufficient training influences nurses’ PC knowledge. Therefore, it is recommended to integrate PC into nursing curricula and in-service training, and that it should address both the basic and advanced principles of PC.

## Figures and Tables

**Table 1 healthcare-11-01781-t001:** Characteristics of Participants (N = 85).

Characteristics	Frequency (*n*)	Percentage (%)
Gender		
Male	49	57.6
Female	36	42.4
Age by year, mean [SD]	31.3 [6.5]	
25 or less	15	17.6
26–30	34	40
More than 30	36	42.4
Educational level		
Diploma	4	4.7
Bachelor	72	84.7
Postgraduate	9	10.6
Position		
Head nurse	8	9.4
Registered nurse	67	78.8
Practical nurse	10	11.8
Total years of working experience	7.9 ± 6.3	
Five or less	34	40
6–10	28	32.9
More than 10	23	27.1
Years of experience in the ICCU	5.4 ± 3.0	
Two or less	32	37.6
3–6	26	30.6
More than 6	27	31.8
History of PC education		
Yes	33	38.8
No	52	61.2

ICCU: intensive coronary care unit; SD: standard deviation; PC: palliative care.

**Table 2 healthcare-11-01781-t002:** Responses of nurses to palliative care knowledge (N = 85).

Items	Correct Answer *n* (%)	Incorrect Answer/Did Not Know *n* (%)
Q1. Palliative care is appropriate only in situations where there is evidence of a downhill trajectory or deterioration. (F)	26 (30.6)	59 (69.4)
Q2. Palliative care is exclusively for people who are in the last six months of life. (F)	20 (23.5)	65 (76.5)
Q3. Palliative care is specifically for people with cancer. (F)	18 (21.2)	67 (78.8)
Q4. Morphine is the standard used to compare the analgesic effect of other opioids. (T)	45 (52.9)	40 (47.1)
Q5. Palliative care is given only to dying patients. (F)	22 (25.9)	63 (74.1)
Q6. Long-term use of opioids can often induce addiction. (T)	64 (75.3)	21 (24.7)
Q7. Some dying patients will require continuous sedation to alleviate suffering. (T)	45 (52.8)	40 (47.2)
Q8. Use of opioids does not influence survival time (T)	48 (56.5)	37 (43.5)
Q9. It is crucial for family members to remain at the bedside until death occurs. (F)	15 (17.6)	70 (82.4)
Q10. The provision of palliative care requires emotional detachment. (F)	44 (51.8)	41 (48.2)
Q11. In palliative nursing care, using placebos is appropriate in treating some types of pain. (T)	18 (21.2)	67 (78.8)
Q12. Men generally reconcile their grief more quickly than women. (F)	19 (22.4)	66 (77.6)
Q13. Terminally ill patients should be encouraged to have hope against all odds. (T)	65 (76.5)	20 (23.5)
Q14. Palliative care is limited to those diagnosed with chronic diseases (Updated definition of 2022). (F)	24 (28.2)	61 (71.8)
Q15. Palliative care is limited to the use of palliative medications only (Updated definition of 2022). (F)	18 (21.2)	67 (78.8)
Total knowledge correct rate 41.4%		

T: the answer to the question is “true”; F: the answer to the question is “false”.

**Table 3 healthcare-11-01781-t003:** Nurses needs about palliative care training (N = 85).

Items	Yes *n* (%)	No *n* (%)
Did you receive training in palliative care?	24 (28.2)	61 (71.8)
Did you receive a pain management program?	21 (24.7)	64 (75.3)
Did you receive a symptoms management program?	23 (27.1)	62 (72.9)
Did you receive a communications skills program (e.g., goals of care discussion, family meetings, breaking bad news)?	27 (31.8)	58 (68.2)
Did you receive ethics issues at the end-of-life program?	19 (22.4)	66 (77.6)
Do you wish to take symptoms management courses?	54 (63.5)	31 (36.5)
Do you wish to take moral courses to help in palliative care?	67 (78.8)	18 (21.2)
Do you wish to take spiritual courses to help in palliative care?	66 (77.6)	19 (22.4)
Do you wish to take communications skills courses (e.g., goals of care discussion, family meetings, breaking bad news)?	68 (80.0)	17 (20)
Do you wish to take courses in support of caring for dying patients?	65 (76.5)	20 (23.5)
Do you wish to take psychological support courses for dying patients?	67 (78.8)	18 (21.2)
Do you wish to take family support courses?	70 (82.4)	15 (17.6)

**Table 4 healthcare-11-01781-t004:** Palliative care services available in the hospital (N = 85).

Services Should be Available at the Hospital	*n* (%)
Guidelines for PC	72 (85.0)
Discharge planning for patients with chronic illness and patients’ needs for palliative care	71 (84.0)
Quality improvement for pain management program	71 (84.0)
Palliative care team or consult service	71 (83.8)
Pain management consultative team	71 (83.6)
The psychological support team for patients’ needs	70 (83.0)
The contractual relationship with a hospice or home care program	70 (82.8)
Program for staff support in caring for dying patients	70 (82.8)
Quality improvement for symptom management program	70 (82.6)
Palliative care unit	69 (82.2)
Quality improvement for the end-of-life-care program	69 (81.2)
Ethics committees should be available in the hospitals	69 (79.8)

**Table 5 healthcare-11-01781-t005:** Ranking of the major barriers to the integration palliative care.

Items	*n* (%)
Lack of knowledge about palliative care by healthcare professionals	62 (72.9)
Personnel shortages/time constraints	57 (67.1)
Lack of designated beds for palliative care services	51 (60.0)
Lack or inadequacy of written policies and procedures about palliative care	47 (55.3)
Improper communication among interdisciplinary team	44 (51.8)
Patients’/families’ avoidance of issues around dying	36 (42.4)
Professionals’ fear of causing addiction by administering pain medications	32 (37.6)
Lack of availability of medications/opioids (narcotics)	29 (34.1)
Professionals’ discomfort with death	25 (29.4)

**Table 6 healthcare-11-01781-t006:** Multiple linear regression for factors associated with nurses’ knowledge about palliative care.

Variable	β	SE	95% CI	Wald	*p* Value
Gender					
Male	0.273	0.4443	−0.59–1.14	0.379	0.538
Female	Ref	-	-	-	-
Age					
25 or less	0.223	0.597	−0.942–1.38	0.141	0.708
More than 26	Ref	-	-	-	-
Educational level					
Diploma	−0.287	1.025	−2.29–1.722	0.078	0.779
Bachelor and high	Ref	-	-	-	-
Total years of working experience	0.062	0.0389	−0.014–1.38	2.566	0.109
History of palliative care education					
Yes	0.202	0.469	−0.71–1.12	0.185	0.667
No	Ref	-	-	-	-

## Data Availability

The data presented in this study are available on request from the corresponding author.

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
