# Peer review of "Towards the Provision of Palliative Care Services in the Intensive Coronary Care Units: Nurses’ Knowledge, Training Needs, and Related-Barriers"

_healthcare, 2023, doi:10.3390/healthcare11121781_

Round 1
Reviewer 1 Report
Thank you for submitting your manuscript.
This manuscript requires attention to several issues.
#1 One of the issues is that the Introduction lacks a clear hypothesis, and it is recommended that you include a hypothesis in the Introduction.
#2 Instead of using the STROBE guideline, it is suggested that you consider using either the SURGE (Survey Reporting Guideline) or a consensus-based checklist for reporting survey studies. Please verify and revise accordingly.
#3 Did the Institutional Review Board approve the study?
If not, please provide a reason for not obtaining IRB approval for this study.
#4 In the methodology section, primary and secondary outcomes should be clearly defined for an appropriate sample size calculation. The sample size calculation cannot be conducted if these outcomes are not defined. Please add these definitions to the methodology section.
#5 Although the multiple imputation method was mentioned in the Data Analysis section, it does not appear to have been used in this study. If this method was not used, please remove it and specify the status of missing data (e.g., "no missing data"). If the multiple imputation method was used, please provide details about the number of datasets and the software used (e.g., mice, with, pool).
#6 Regarding the sample size calculation, we noticed that you cited reference #24, which is a book, and the formula provided could not be verified. It could not be found despite efforts to locate the formula through other means. We kindly request that you provide the citation of the article that mentions the Thompson equation formula to support the accuracy and validity of the sample size calculation. Failure to do so may jeopardize the acceptance of this manuscript for publication.
#7 The presentation of age information (mean and standard deviation) in Table 1 is not clearly defined and may cause confusion, so please consider revising this presentation.
#8 The use of a generalized linear regression model in analyzing the nurses' knowledge of primary care may be problematic due to the small sample size (approximately 80). Only up to 5 factors may be used in the analysis, and the number of variables for categorical variables should be calculated as (types of categories - 1). For example, if "education level" has three categories, the number of variables should be counted as 2. If you decide to use this analysis, more data should be collected, or the number of factors should be adjusted.
#9 The methodology section of this manuscript requires significant improvement, which may affect the validity of the results. Therefore, it is difficult to provide feedback on the Discussion section now. Please address the issues in the methodology section before proceeding with the Discussion.
Author Response
27th June. 2023
Manuscript ID: healthcare-2374315, entitled "Towards the Provision of Palliative Care Service in the Intensive Coronary Care Units: Nurses Knowledge, Training Needs, and Barriers"
Dear Editor and reviewers,
Thank you for giving us the apportunity to revise our manuscript and to respond to the helpful comments raised by the reviewers. Below is a detailed response to comments and changes made to the manuscript have been highlighted by tack changes.
Response to Reviewer (1)
Thank you for submitting your manuscript. This manuscript requires attention to several issues.
Comment (1) One of the issues is that the Introduction lacks a clear hypothesis, and it is recommended that you include a hypothesis in the Introduction.
Response: Thank you so much for all your comments that have inspired us to improve this version. As clearly mentioned in the method section, this is a descriptive cross-sectional quantitative study. In general, descriptive cross-sectional studies are not designed to test a specific hypothesis, but rather to describe the prevalence or distribution of a particular characteristic or outcome at a particular point in time. In this study, the primary outcome/goal is to assess the knowledge of nurses about palliative care, which is clearly mentioned in P 3, Lines 101–103, as "This study aimed to assess the level of knowledge and needs of palliative care (PC) among nurses towards the provision of PC in intensive coronary care units in the Gaza Strip as well as identify the barriers to the provision of PC services in ICCUs". However, some literature mentions the possibility for a descriptive cross-sectional study to include a hypothesis, and the hypothesis may be focused on describing the association between variables rather than testing a specific cause-and-effect relationship. So to address your suggestion, one more objective has been added: “In addition, it aimed to examine the relationships between nurses’ demographic and professional variables and their knowledge about PC ". (P 3, Lines 103–104). Thus, the null hypothesis has been written as “Our null hypothesis posited that no associations regarding all demographic and pro-fessional variables and nurses' knowledge about PC would be found ". (P 3, Lines 103–105).
Comment (2) Instead of using the STROBE guideline, it is suggested that you consider using either the SURGE (Survey Reporting Guideline) or a consensus-based checklist for reporting survey studies. Please verify and revise accordingly.
Response: According to your recommendation, the SURGE (Survey Reporting Guideline) has been adopted for reporting current study (Grimshaw, 2014). (P 3, Lines 117–119). The manuscript has been revised accordingly to adhere to the SURGE guidelines.
Comment (3) Did the Institutional Review Board approve the study? If not, please provide a reason for not obtaining IRB approval for this study.
Response: Yes an approval was obtained from The Islamic University of Gaza Ethics Committee (Ref No: 07/2021/GSGH). A new sentence has been added in the method section about ethical approval. (P 3, Lines 114–117).
Comment (4) In the methodology section, primary and secondary outcomes should be clearly defined for an appropriate sample size calculation. The sample size calculation cannot be conducted if these outcomes are not defined. Please add these definitions to the methodology section.
Response: More information has been added to the method section about primary and secondary outcomes. Further, a section about sociodemographic and professional variables has been added. (P 4, Lines 172–175).
Comment (5) Although the multiple imputation method was mentioned in the Data Analysis section, it does not appear to have been used in this study. If this method was not used, please remove it and specify the status of missing data (e.g., "no missing data"). If the multiple imputation method was used, please provide details about the number of datasets and the software used (e.g., mice, with, pool).
Response: Thank you so much for your comment. We deleted the multiple imputation method from the "statistical analysis" section, and in the result section, we have clearly highlighted that there was no missing data as "No missing data was reported in the questionnaires that were filled out by the study participants". (P 4, Lines 178–179).
Comment (6) Regarding the sample size calculation, we noticed that you cited reference #24, which is a book, and the formula provided could not be verified. It could not be found despite efforts to locate the formula through other means. We kindly request that you provide the citation of the article that mentions the Thompson equation formula to support the accuracy and validity of the sample size calculation. Failure to do so may jeopardize the acceptance of this manuscript for publication.
Response: Thank you so much for your comments. The Steven K. Thompson equation is one of the recommended formulas that has previously been used by researchers to estimate the sample size when the total population is clearly identified. During the insertion of the citation reference, we have cited by error the Thompson-Wiliam book, which differs from Steven K. Thompson's book. We have updated the correct book citation reference (Thompson, 2012), and we have cited the pages where the equation is written. Furthermore, the sample size calculation formula has been added to the manuscript as “Utilising the Steven K. Thompson equation formula (n = N × p (1-p)/[[N-1 × (d2÷z2)] + p (1-p)] (N = population size, Z = confidence level, d = error proportion, and p = probability) (Thompson, 2012) (Page. 59-60), the required sample size was 83 nurses (Z = 1.96, d = 0.05, and p = 0.5). (P 3, Lines 129–131).
Comment (7) The presentation of age information (mean and standard deviation) in Table 1 is not clearly defined and may cause confusion, so please consider revising this presentation.
Response: The presentation of age information (mean and standard deviation) in Table 1 and in text has been modified to be clear to the readers.
Comment (8) The use of a generalized linear regression model in analyzing the nurses' knowledge of primary care may be problematic due to the small sample size (approximately 80). Only up to 5 factors may be used in the analysis, and the number of variables for categorical variables should be calculated as (types of categories - 1). For example, if "education level" has three categories, the number of variables should be counted as 2. If you decide to use this analysis, more data should be collected, or the number of factors should be adjusted.
Response: Thank you so much for your critical comment. As it is difficult to collect more data, the data has been checked carefully and adjustment on the factors has been made. Please check the updated generalized linear regression model table 6.
Comment (9) The methodology section of this manuscript requires significant improvement, which may affect the validity of the results. Therefore, it is difficult to provide feedback on the Discussion section now. Please address the issues in the methodology section before proceeding with the Discussion.
Response: All of your comments have been addressed, and we hope to check the discussion section and giving us your feedback to improve it if needed.
Additional clarifications
In addition to the above comments, the authors have checked all spelling and grammatical errors. The revised manuscript conforms to the journal style.
We are happy to provide any further clarification if necessary. We hope the above changes have now fully addressed the very useful comments made by the reviewers, and we appreciate their input and time on this, which helped us improve our paper.
Sincerely,
Hammoda Abu-Odah
(on behalf of all authors)
Grimshaw, J. M. (2014). SURGE (The SUrvey Reporting GuidelinE). In Moher D, Altman DG, Wager E, Simera I, & Schulz KF (Eds.), Guidelines for reporting health research: A user's manual. . Chichester, West Sussex, Hoboken, NJ: John Wiley & Sons Ltd; 2014; 206-2013.
Kozlov, E., Carpenter, B. D., & Rodebaugh, T. L. (2017). Development and validation of the Palliative Care Knowledge Scale (PaCKS). Palliat Support Care, 15(5), 524-534. doi:10.1017/s1478951516000997
Thompson, S. (2012). Sampling (3rd ed.). . Canada: John Wiley & Sons, Inc.
Steven K. Thompson equation formula Thompson, S. (2012). Sampling (3rd ed.). . Canada: John Wiley & Sons, (Page. 59-60)

Reviewer 2 Report
This is an interesting report that attempts to identify the knowledge, training needs, and barriers to the provision of palliative care among nurses working in ICCUs in the Gaza Strip of Palestine. However, as a scientific article, it is not well described and could be improved.
1. Abstract (Page 1)
In the abstract, you stated that the data were collected by validated instruments. However, the "Knowledge about Palliative Care" was measured by the questionnaire which was created by yourself with reference to the existing scales (the PCQN and the PCKT). The measurement you used had been not tested for reliability and validity, had it?
2. Material and Methods (page 3)
1) sample size calculation
In describing the sample size calculation, the paper you cited is a book for estimating the population of rare animals, how did you use it in this sample size calculation? Please specify the sample size calculation formula used in this study.
2) Data analysis
What of the generalized linear regression analysis was used? Was it a linear regression analysis?
It does not specify whether it was a single regression analysis or a multiple regression analysis, so I don't know. If a linear regression analysis was used, could a normal distribution be assumed for the distribution of the knowledge measures? Since it says that univariate analysis was conducted by selecting all variables with p-values less than 0.05, it would be good to show the results of that analysis. However, if the results in Table 6 were the results of a multivariate analysis, it seems unlikely that all of each variable entered into the multivariated analysis had a p-value less than 0.05 in the univariate analysis.
3) Lack of description of the recruitment procedures
Please specify the specific procedures for explaining the research participation and obtaining consent. Please also specify the procedures for distributing and collecting the survey forms.
Author Response
27th June. 2023
Manuscript ID: healthcare-2374315, entitled "Towards the Provision of Palliative Care Service in the Intensive Coronary Care Units: Nurses Knowledge, Training Needs, and Barriers"
Dear Editor and reviewers,
Thank you for giving us the apportunity to revise our manuscript and to respond to the helpful comments raised by the reviewers. Below is a detailed response to comments and changes made to the manuscript have been highlighted by tack changes.
Response to Reviewer (2)
This is an interesting report that attempts to identify the knowledge, training needs, and barriers to the provision of palliative care among nurses working in ICCUs in the Gaza Strip of Palestine. However, as a scientific article, it is not well described and could be improved.
Comment (1) Abstract (Page 1) In the abstract, you stated that the data were collected by validated instruments. However, the "Knowledge about Palliative Care" was measured by the questionnaire which was created by yourself with reference to the existing scales (the PCQN and the PCKT). The measurement you used had been not tested for reliability and validity, had it?
Response: Thank you so much for highlighting this comment. Modification has been added to the abstract confirming that the developed instrument was based on PCQN and PCKT. Yes, the instrument had not been tested for reliability and validity, which considered as one of the limitations of this study. A new sentence has been added to the limitation section about this point. (P 10, Lines 319–323).
Material and Methods (page 3)
Comment (2) 1) sample size calculation: In describing the sample size calculation, the paper you cited is a book for estimating the population of rare animals, how did you use it in this sample size calculation? Please specify the sample size calculation formula used in this study.
Response: Thank you so much for your comments. The Steven K. Thompson equation is one of the recommended formulas that has previously been used by researchers to estimate the sample size when the total population is clearly identified. During the insertion of the citation reference, we have cited by error the Thompson-Wiliam book, which differs from Steven K. Thompson's book. We have updated the correct book citation reference (Thompson, 2012), and we have cited the pages where the equation is written. Furthermore, the sample size calculation formula has been added to the manuscript as “Utilising the Steven K. Thompson equation formula (n = N × p (1-p)/[[N-1 × (d2÷z2)] + p (1-p)] (N = population size, Z = confidence level, d = error proportion, and p = probability) (Thompson, 2012) (Page. 59-60), the required sample size was 83 nurses (Z = 1.96, d = 0.05, and p = 0.5). (P 3, Lines 129–131).
Comment (3) 2) Data analysis: What of the generalized linear regression analysis was used? Was it a linear regression analysis? It does not specify whether it was a single regression analysis or a multiple regression analysis, so I don't know. If a linear regression analysis was used, could a normal distribution be assumed for the distribution of the knowledge measures? If the results in Table 6 were the results of a multivariate analysis, it seems unlikely that all of each variable entered into the multivariated analysis had a p-value less than 0.05 in the univariate analysis.
Response: Linear regression is part of the generalized linear model. The generalized linear model is a generalization of linear regression. We adopted the Generalized linear model as it is more flexible, which model allows us to include both continuous or categorical variables in the model. In our study, we performed multiple linear regression as we have more than two outcome variables. To make this clear, " generalized linear model " has replaced in the manuscript by " Multiple Linear Regression ". The data were asseesed for normality by using the Kolmogorov–Smirnov test and the Shapiro–Wilk test and the graphical interpretation (a histogram) to determine normality. The data were normality distributed. Additional information has been added to the "statistical method" section. Furthermore, according to another reviewer suggestion, Table 6 has been checked and updated.
Comment (4) 3) Lack of description of the recruitment procedures. Please specify the specific procedures for explaining the research participation and obtaining consent. Please also specify the procedures for distributing and collecting the survey forms.
Response: A new subsection (Section 2.3) has been added to the method section presenting the recruitment procedures. (P 3, Lines 133–143).
Additional clarifications
In addition to the above comments, the authors have checked all spelling and grammatical errors. The revised manuscript conforms to the journal style.
We are happy to provide any further clarification if necessary. We hope the above changes have now fully addressed the very useful comments made by the reviewers, and we appreciate their input and time on this, which helped us improve our paper.
Sincerely,
Hammoda Abu-Odah
(on behalf of all authors)
Grimshaw, J. M. (2014). SURGE (The SUrvey Reporting GuidelinE). In Moher D, Altman DG, Wager E, Simera I, & Schulz KF (Eds.), Guidelines for reporting health research: A user's manual. . Chichester, West Sussex, Hoboken, NJ: John Wiley & Sons Ltd; 2014; 206-2013.
Kozlov, E., Carpenter, B. D., & Rodebaugh, T. L. (2017). Development and validation of the Palliative Care Knowledge Scale (PaCKS). Palliat Support Care, 15(5), 524-534. doi:10.1017/s1478951516000997
Thompson, S. (2012). Sampling (3rd ed.). . Canada: John Wiley & Sons, Inc.
Steven K. Thompson equation formula Thompson, S. (2012). Sampling (3rd ed.). . Canada: John Wiley & Sons, (Page. 59-60)

Reviewer 3 Report
The subject is original.
The title is compatible with the study and method.
The method is explained in detail. The authors explained that they made some selections from the questions in the questionnaires in the study. However, they did not specify which questions were removed and it is uncertain how these will affect the study (biased?).
The results are written clearly and precisely.
In the references section, there are 8 publications (sources 3, 13, 14, 16, 20, 21, 22, 37) of the researcher named Hammoda Abu-Odah. Sources 34 and 35 are duplicated, correction is required.
Ethics committee approval has been reported.
Author Response
27th June. 2023
Manuscript ID: healthcare-2374315, entitled "Towards the Provision of Palliative Care Service in the Intensive Coronary Care Units: Nurses Knowledge, Training Needs, and Barriers"
Dear Editor and reviewers,
Thank you for giving us the apportunity to revise our manuscript and to respond to the helpful comments raised by the reviewers. Below is a detailed response to comments and changes made to the manuscript have been highlighted by tack changes.
Response to Reviewer (3)
Comment (1) The subject is original. The title is compatible with the study and method. The method is explained in detail. The authors explained that they made some selections from the questions in the questionnaires in the study. However, they did not specify which questions were removed and it is uncertain how these will affect the study (biased?).
Response: Thank you so much for your comments. We acknowledge that excluding some items from the validated original questionnaires is not appropriate; however, the deleted items were based on the suggestion of the Palestinian experts’ panel, as these items were not practically applicable to the Palestinian context. From the Palliative Care Knowledge Test (PCKT), some of items related to pain (item 4 and 6), dyspnoea (item 9-12), psychiatric problems (item 13 and 14), and gastrointestinal problems (item 17-20) domains were excluded. for instance, item 4 in the pain domain entitled “When cancer pain is mild, pentazocine should be used more often than an opioid” was excluded because pentazocine was not utilised in Gaza hospitals. On the other hand, the panel recommended adding some items, including “PC is exclusively for people who are in the last six months of life” (Kozlov, Carpenter, & Rodebaugh, 2017), 2) and “PC is specifically for people with cancer” (Kozlov et al., 2017). (P 4, Lines 152–160). We have highlighted the exclusion of items as a limitation of the study and mentioned it clearly in the limitation section: "In addition, the exclusion of some items in the PCQN and PCKT instruments based on Palestinian experts’ panel opinion makes it difficult to compare our study findings with other studies in different populations". (Page 10, Lines 319-323).
Comment (2) The results are written clearly and precisely.
Response: Thank you so much for your kindness.
Comment (3) In the references section, there are 8 publications (sources 3, 13, 14, 16, 20, 21, 22, 37) of the researcher named Hammoda Abu-Odah. Sources 34 and 35 are duplicated, correction is required.
Response: This study is based on the work that has been done by Hammoda Abu-Odah, one of the co-authors, who contributes to the development of palliative care in the Gaza Strip. However, the reference lists have been checked and adjusted.
Additional clarifications
In addition to the above comments, the authors have checked all spelling and grammatical errors. The revised manuscript conforms to the journal style.
We are happy to provide any further clarification if necessary. We hope the above changes have now fully addressed the very useful comments made by the reviewers, and we appreciate their input and time on this, which helped us improve our paper.
Sincerely,
Hammoda Abu-Odah
(on behalf of all authors)
Grimshaw, J. M. (2014). SURGE (The SUrvey Reporting GuidelinE). In Moher D, Altman DG, Wager E, Simera I, & Schulz KF (Eds.), Guidelines for reporting health research: A user's manual. . Chichester, West Sussex, Hoboken, NJ: John Wiley & Sons Ltd; 2014; 206-2013.
Kozlov, E., Carpenter, B. D., & Rodebaugh, T. L. (2017). Development and validation of the Palliative Care Knowledge Scale (PaCKS). Palliat Support Care, 15(5), 524-534. doi:10.1017/s1478951516000997
Thompson, S. (2012). Sampling (3rd ed.). . Canada: John Wiley & Sons, Inc.
Steven K. Thompson equation formula Thompson, S. (2012). Sampling (3rd ed.). . Canada: John Wiley & Sons, (Page. 59-60)

Reviewer 4 Report
Although the idea of the research is not a novel one, but being first time performed in GAZA strip.
PC in middle east is an emerging concept being established in only few centers in Kuwait, KSA, and Egypt, thus this research is an important initial step to identify the needs and barriers to PC training In Palestinian health care staff.
the main strengths of the manuscripts are:
1. To my knowledge this is the first research address these objectives among Arab Nurses
2. The methodology is clearly explained.
3. The basic sections were adequate with good use of subheadings.
4. The writing order was logic and clear.
5. The writer cited sources adequately and appropriately.
however, the exclusion of some questions in The Palliative Care Quiz for Nursing (PCQN) and the Palliative Care Knowledge Test (PCKT) based on experts panel opinion was not required as using the standardized widely used version of the validated tools would help in further research comparing your result with other studies in different populations. this may worth mentioning in the limitations section.
Author Response
27th June. 2023
Manuscript ID: healthcare-2374315, entitled "Towards the Provision of Palliative Care Service in the Intensive Coronary Care Units: Nurses Knowledge, Training Needs, and Barriers"
Dear Editor and reviewers,
Thank you for giving us the apportunity to revise our manuscript and to respond to the helpful comments raised by the reviewers. Below is a detailed response to comments and changes made to the manuscript have been highlighted by tack changes.
Response to Reviewer (4)
Although the idea of the research is not a novel one, but being first time performed in GAZA strip. PC in Middle East is an emerging concept being established in only few centers in Kuwait, KSA, and Egypt, thus this research is an important initial step to identify the needs and barriers to PC training In Palestinian health care staff. The main strengths of the manuscripts are: 1. To my knowledge this is the first research address these objectives among Arab Nurses, 2. The methodology is clearly explained, 3. The basic sections were adequate with good use of subheadings, 4. The writing order was logic and clear, and 5. The writer cited sources adequately and appropriately.
Comment (1) However, the exclusion of some questions in The Palliative Care Quiz for Nursing (PCQN) and the Palliative Care Knowledge Test (PCKT) based on experts panel opinion was not required as using the standardized widely used version of the validated tools would help in further research comparing your result with other studies in different populations. This may worth mentioning in the limitations section.
Response: Thank you so much for your comments. We acknowledge that excluding some items from the validated original questionnaires is not appropriate; however, the deleted items were based on the suggestion of the Palestinian experts’ panel, who recommended to focus more on philosophy items related PC such as definition, and the group of people who can benefits from PC. We have highlighted the exclusion of items as a limitation of the study and mentioned it clearly in the limitation section: "In addition, the exclusion of some items in the PCQN and PCKT instruments based on Palestinian experts’ panel opinion makes it difficult to compare our study findings with other studies in different populations". (Page 10, Lines 319-323).
Additional clarifications
In addition to the above comments, the authors have checked all spelling and grammatical errors. The revised manuscript conforms to the journal style.
We are happy to provide any further clarification if necessary. We hope the above changes have now fully addressed the very useful comments made by the reviewers, and we appreciate their input and time on this, which helped us improve our paper.
Sincerely,
Hammoda Abu-Odah
(on behalf of all authors)
Grimshaw, J. M. (2014). SURGE (The SUrvey Reporting GuidelinE). In Moher D, Altman DG, Wager E, Simera I, & Schulz KF (Eds.), Guidelines for reporting health research: A user's manual. . Chichester, West Sussex, Hoboken, NJ: John Wiley & Sons Ltd; 2014; 206-2013.
Kozlov, E., Carpenter, B. D., & Rodebaugh, T. L. (2017). Development and validation of the Palliative Care Knowledge Scale (PaCKS). Palliat Support Care, 15(5), 524-534. doi:10.1017/s1478951516000997
Thompson, S. (2012). Sampling (3rd ed.). . Canada: John Wiley & Sons, Inc.
Steven K. Thompson equation formula Thompson, S. (2012). Sampling (3rd ed.). . Canada: John Wiley & Sons, (Page. 59-60)

Reviewer 5 Report
Thank you for an interesting manuscript.
I have some comments and suggestions for improvement:
Abstract: The validated instruments should ne named in the abstracts.
Introduction: The introduction is too long. Please shorten the introduction.
Materials and methods: I miss pilot-testing of the questionnaire. Has there been done any testing before?
Limitations: The limitations need to be expanded. The lacking of pilot-testing is one limitation.
Author Response
27th June. 2023
Manuscript ID: healthcare-2374315, entitled "Towards the Provision of Palliative Care Service in the Intensive Coronary Care Units: Nurses Knowledge, Training Needs, and Barriers"
Dear Editor and reviewers,
Thank you for giving us the apportunity to revise our manuscript and to respond to the helpful comments raised by the reviewers. Below is a detailed response to comments and changes made to the manuscript have been highlighted by tack changes.
Response to Reviewer (5)
Thank you for an interesting manuscript. I have some comments and suggestions for improvement:
Comment (1) Abstract: The validated instruments should ne named in the abstracts.
Response: Thank you so much for your comments. New information has been added to the abstract section about the instruments used.
Comment (2) Introduction: The introduction is too long. Please shorten the introduction.
Response: The introduction has been shorten as your suggestion.
Comment (3) Materials and methods: I miss pilot-testing of the questionnaire. Has there been done any testing before?
Response: Piloting has not been done, which is one of the limitation of the study.
Comment (4) Limitations: The limitations need to be expanded. The lacking of pilot-testing is one limitation.
Response: Lack of piloting as a limitation has been added in the limitation section as “Furthermore, the lack of pilot test make it is difficult to determine the validity of the study instruments’ and identify their weakness”. ". (Page 10, Lines 319-323).
Additional clarifications
In addition to the above comments, the authors have checked all spelling and grammatical errors. The revised manuscript conforms to the journal style.
We are happy to provide any further clarification if necessary. We hope the above changes have now fully addressed the very useful comments made by the reviewers, and we appreciate their input and time on this, which helped us improve our paper.
Sincerely,
Hammoda Abu-Odah
(on behalf of all authors)
Grimshaw, J. M. (2014). SURGE (The SUrvey Reporting GuidelinE). In Moher D, Altman DG, Wager E, Simera I, & Schulz KF (Eds.), Guidelines for reporting health research: A user's manual. . Chichester, West Sussex, Hoboken, NJ: John Wiley & Sons Ltd; 2014; 206-2013.
Kozlov, E., Carpenter, B. D., & Rodebaugh, T. L. (2017). Development and validation of the Palliative Care Knowledge Scale (PaCKS). Palliat Support Care, 15(5), 524-534. doi:10.1017/s1478951516000997
Thompson, S. (2012). Sampling (3rd ed.). . Canada: John Wiley & Sons, Inc.
Steven K. Thompson equation formula Thompson, S. (2012). Sampling (3rd ed.). . Canada: John Wiley & Sons, (Page. 59-60)

Round 2
Reviewer 2 Report
Material and Methods 2.2 sample calculation and participant selection(page 3)
The formula used as sample size calculation based on Thompson's literature is a calculation method based on the accuracy of point estimates of proportions. Was the proportion the main outcome of this study?
Sample size is one of the key indicators to ensure accuracy and representativeness, but since the population is quite small (105 participants), the presence of background bias between those who were eligible at the time of recruitment and those who were not (e.g., one facility was not included in the participants) is also important.
Author Response
10th June. 2023
Manuscript ID: healthcare-2374315, entitled "Towards the Provision of Palliative Care Service in the Intensive Coronary Care Units: Nurses Knowledge, Training Needs, and Barriers"
Dear Editor and reviewer,
Thank you for again for allowing us to revise our manuscript and to respond to the helpful comments raised by the reviewer. Below is a detailed response to comments and changes made to the manuscript have been highlighted by tack changes.
Response to Reviewer (2)
Material and Methods 2.2 sample calculation and participant selection(page 3)
Comment : The formula used as sample size calculastion based on Thompson's literature is a calculation method based on the accuracy of point estimates of proportions. Was the proportion the main outcome of this study? Sample size is one of the key indicators to ensure accuracy and representativeness, but since the population is quite small (105 participants), the presence of background bias between those who were eligible at the time of recruitment and those who were not (e.g., one facility was not included in the participants) is also important.
Response: Thank you so much for your crucial comments that have been raised during the reviewing process. All of your comments have been given the highest priority. In our study, the primary outcome is "the level of PC knowledge", which is a continuous variable. So we agree with you that this formula is not suitable, as it applies when the variable used is binary. After a lengthy discussion with the statistician and based on his advice, we have utilised Epi-Info-7 StatCalc programme for calculation sample size. A modification has been done as "In total, there were 105 nurses working at the aforementioned hospitals. The sample was calculated using the Epi-Info-7 StatCalc programme with the following assumptions: a statistical power of 80%, a confidence level of 95%, an expected frequency/prevalence 50%, and a margin of error of 5%, a minimum sample size of 82 participants was included. In this study, 85 nurses participated and were willing to complete the survey". (Page 3, Lines 118-125).
An acknowledgments section has been added as “The authors would like to thank all nurses who participated in this study. We thank the statistician Dr Ayman Abu Mostafa for his advice during the analysis of the data”.
In addition, we acknowledge that the small population may influence on selection bias of the participants, which is one limitation of our study. A sentence has been added in the limitation section as “Considering that the population in this study is quite small, the risk of background bias may occur, which may affect the accuracy and representativeness of the findings”. (Page 9, Lines 309-311).
We are happy to provide any further clarification if necessary. We hope the above changes have now fully addressed the very useful comments made by the reviewer, and we appreciate their input and time on this, which helped us improve our paper.
Sincerely,
Hammoda Abu-Odah
(on behalf of all authors)
